# Transcriptomic analysis of a wild and a cultivated varieties of *Capsicum annuum* over fruit development and ripening

**Fernando G. Razo-Mendivil** [1], **Fernando Hernandez-Godínez**[2], **Corina Hayano-Kanashiro**[1]*, **Octavio Martínez**[2]*

**1** Departamento de Investigaciones Científicas y Tecnológicas de la Universidad de Sonora, Universidad de Sonora, Hermosillo, México, **2** Unidad de Genómica Avanzada (UGA-Langebio), Centro de Investigación y de Estudios Avanzados del Instituto Politécnico Nacional (Cinvestav), Irapuato, México

\* angela.hayano@unison.mx (CHK); octavio.martinez@cinvestav.mx (OM)

## Abstract

Chili pepper (*Capsicum annuum*) is one of the most important crops worldwide. Its fruits contain metabolites produced over the maturation process like capsaicinoids and carotenoids. This metabolic process produces internal changes in flavor, color, texture, and aroma in fruits to make them more attractive for seed dispersal organisms. The chiltepin (*C. annuum* L. *var. glabriusculum*) is a wild variety of the *C. annuum* L. species that is considered a source of genetic resources that could be used to improve the current chili crops. In this study, we performed a transcriptomic analysis on two fruit maturation stages: immature stage (green fruit) and mature stage (red fruit) of a wild and a cultivated pepper variety. We found 19,811 genes expressed, and 1,008 genes differentially expressed (DEGs) in at least one of the five contrast used; 730 DEGs were found only in one contrast, and most DEGs in all contrasts were downregulated. GO enrichment analysis showed that the majority of DEGs are related to stress responses. KEGG enrichment analysis detected differences in expression patterns in metabolic pathways related to phenylpropanoid biosynthesis, secondary metabolites, plant hormone signal transduction, carotenoid biosynthesis and sesquiterpenoid and triterpenoid biosynthesis. We selected 105 tomato fruit ripening-related genes, and found 53 pepper homologs differentially expressed related to shape, size, and secondary metabolite biosynthesis. According to the transcriptome analysis, the two peppers showed very similar gene expression patterns; differences in expression patterns of genes related to shape, size, ethylene and secondary metabolites biosynthesis suggest that changes produced by domestication of chilli pepper could be very specific to the expression of genes related to traits desired in commercial fruits.

## Introduction

Domestication is a co-evolutionary process produced by the relationship between a domesticator and a domesticate [1]. The domestication process starts with the consumption of wild

**Data Availability Statement:** Transcriptomic information has been deposited in the NCBI Gene Expression Omnibus (GEO) database with

accession number GSE171889 (https://www.ncbi.nlm.nih.gov/geo/query/acc.cgi?acc=GSE171889). All other relevant data are in the paper and its Supporting information files.

**Funding:** This work was supported in part by COFUPRO (A/GTO/RGAG-2014-076-Consorcio de Fundaciones PRODUCE). FR-M acknowledges the Mexican Council of Science and Technology (CONACyT) for supporting with a PhD scholarship (261122) during the development of the project. The funders had no role in study design, data collection and analysis, decision to publish, or preparation of the manuscript.

**Competing interests:** The authors have declared that no competing interests exist.

plants that eventually leads to their cultivation, assuming control over all aspects of its life cycle. Domestication gives rise to new species or differentiated populations and ends with the selection of the desired traits selected by humans for their survival [1, 2]. Domesticated plants display a domestication syndrome which consists of a significant increase in the size of their harvest parts, loss of dispersal ability, seed dormancy, and protection against herbivory. They become reliant on humans for their growth and reproduction, making them less capable to survive in the wild [2].

Chili pepper (*Capsicum sp.*) is one of the most important crops in the world [3]. It possesses agricultural and economic importance for Latin American, Asian and African countries [4, 5]. There, pepper fruits are used for human consumption, as natural coloring agents [6], food preservatives [7], an ingredient in pharmaceuticals, and as the key ingredient of self-defense sprays [8].

The *Capsicum* genus belongs to the *Solanaceae* family, native to southern North America and northern South America [9, 10]. The *Solanaceae* family includes other important crops like potato (*Solanum tuberosum*), tomato (*Solanum lycopersicum*), tobacco (*Nicotiana tabacum*), eggplant (*Solanum melongena*), and ornamentals like petunia (*Petunia sp.*). A different number of species have been reported for the *Capsicum* genus depending on the author, from 25 to 36 species [11]. Only five species are domesticated: *C. annuum* L. (Mexico and the northern part of Central America), *C. baccatum* L. (Peru and Bolivia), *C. chinense* Jacq. (West Indies), *C. frutescens* L. (mainly in the Caribbean and South America), and *C. pubescens* Ruiz & Pavon (found at higher altitudes in the Andes) [12–14].

*C. annuum* was domesticated in Mexico around 6,000 years ago, where it has been cultivated and consumed for hundreds of years [5, 15–17]. *C. annuum* is the most important and widely grown around the world [3, 18]. The fruits from different cultivars of this species display a lot of morphological differences in shape, size, and levels of pungency ranging from sweet to hot [19, 20].

Chiltepin (*C. annuum* L. var. *grabliusculum* (Dunal) Heiser & Pickersgill) is a wild variety of the *C. annuum* species distributed from the southwestern United States, Mexico, Central America to Colombia [21]. In Mexico, Chiltepin grows along the Pacific and Mexican Gulf coasts, from Sonora to Chiapas and from Tamaulipas to Quintana Roo [21–24]. This variety is the most likely progenitor of the domesticated *C. annuum* cultivars [25] and is considered an important genetic resource of primary genes that can be used for pepper crop improvement against drought, plagues, and diseases [26].

Pepper fruits are an excellent source of compounds such as carotenoids responsible for their diverse and attractive colors [27, 28] vitamins E, C, B complex, and provitamin A which bring health properties; polyphenol compounds such as flavonoids and cinnamic acid derivatives with anticancer activity; carbohydrates, sugars, calcium, magnesium [12, 29–31], as well as capsaicinoids, alkaloid compounds that confer peppers their characteristic pungency [12, 32, 33].

Tomato, a close relative of pepper, has been used as a model to study fleshy fruit development and ripening. Tomato fruit develops from the ovary in four stages: (1) flower fertilization, (2) proliferation of pericarp cells, (3) fruit growth due to cell expansion and endoreduplication ends with a mature green fruit (MG stage) with the fruit final size, and (4) ripening [34]. Fruit ripening is a biochemical process with dramatic metabolic changes [35] that produces the organoleptic properties characteristic of the pepper fruits. The ripening process has evolved to produce fruits attractive to seed dispersal organisms by activating metabolic pathways that biosynthesize pigments, sugars, acids, and volatile compounds (aroma associated) as well as promoting tissue softening and degradation for easy seed release [36].

Because of the economic and agricultural importance of chili peppers, and being a species used as a model organism for classical and molecular genetic analyses [37–39], many studies have been focused on analyzing the transcriptome of *Capsicum sp.* as a source of basic and applied knowledge. For example, transcriptome analysis has been used to identify genes involved in the biosynthesis of capsaicinoids [32, 40], and in response to different types of stress [41–43].

Previous transcriptome analysis of the genes involved in the chili pepper response by the geminivirus infection identified 309 differentially expressed genes between a control (healthy) and an infected tissue [41]. A study of the transcriptome from seedlings of heat-susceptible (S590) and heat-tolerant (R597) chili pepper varieties under heat stress reported 3,799 and 4,010 genes differentially expressed from R597 and S590, respectively, and 35 genes were confirmed to be involved in stress response [42]. A global transcriptomic analysis of chili pepper plants under different types of biotic/abiotic stress revealed that salicylic acid (SA) related genes activation may be affected by abiotic stress, methyl jasmonate (MeJA) and ethylene (ET) responsive genes may be affected by biotic stress, and abscisic acid-related genes may be affected by both biotic and abiotic stress [43]. A study of the transcriptome of chili pepper (*Capsicum annuum* L.; 'tampiqueño 74') during fruit maturation was carried out, sampling fruits at 10, 20, 40, and 60 days after anthesis (DAA). The most drastic changes in gene expression were reported from 10 to 20 DAA, and from 40 to 60 DAA; showing that the last interval is characterized by a downfall in genes expressed, indicating the end of the maturation process and the beginning of the senescence [44].

In the present study, we analyzed a wild and a cultivated pepper variety over two developmental stages: green (immature stage) and red fruit (mature stage) through Illumina-based transcriptome analysis. We found that the fruit ripening process is well conserved in *C. annuum* L. cv. 'tampiqueño 74' and *Capsicum annuum var. glabriusculum* 'Chiltepin'; and morphological variability may be related to differences in the expression pattern of a small set of genes.

## Results

### RNASeq summary

Transcriptomic analysis of the wild Chiltepin (CH; *Capsicum annuum var. glabriusculum*) and the cultivated Serrano Tampiqueño (ST; *Capsicum annuum* L. cv. 'Tampiqueño 74') was performed at two stages (Fig 1A), the immature stage at 20 days after anthesis (DAA) for both CH and ST, and at mature state at 60 DAA for ST and 68 DAA for CH. All four samples of CH were purified and sequenced, yielding 260,680,166 raw reads (S1 Fig). Raw reads were trimmed of sequencing adaptors, and low-quality reads were filtered out. After filtering, 84G of data was produced, with 250,094,380 reads between 50 and 150 bp, and a quality Phred score $\geq$ 26 (S1 Table, S2 Fig). Clean reads were mapped against the reference genome (*Capsicum annuum* L. cv. 'Zunla-1') with a minimum unique alignment rate of 80% in all samples (S1 Table). Mapped reads were used to estimate gene abundances, which were used in subsequent differential expression analysis. Data of the cultivated variety (Serrano tampiqueño) was obtained as raw counts table from an ongoing research project [45].

We found 19,811 genes expressed in at least one of the CH and ST samples. The number of expressed genes per condition was very similar in all cases, ranging from 18,590 in ST 60DAA to 19,525 in CH 20DAA. Fig 1B shows a Venn diagram of the expressed genes at each condition and their intersections. 18,067 (91.2%) genes were expressed in all conditions. 443 genes (2.24%) were expressed only at 20DAA, 6 only in ST and 31 in CH. 43 genes (0.22%) were

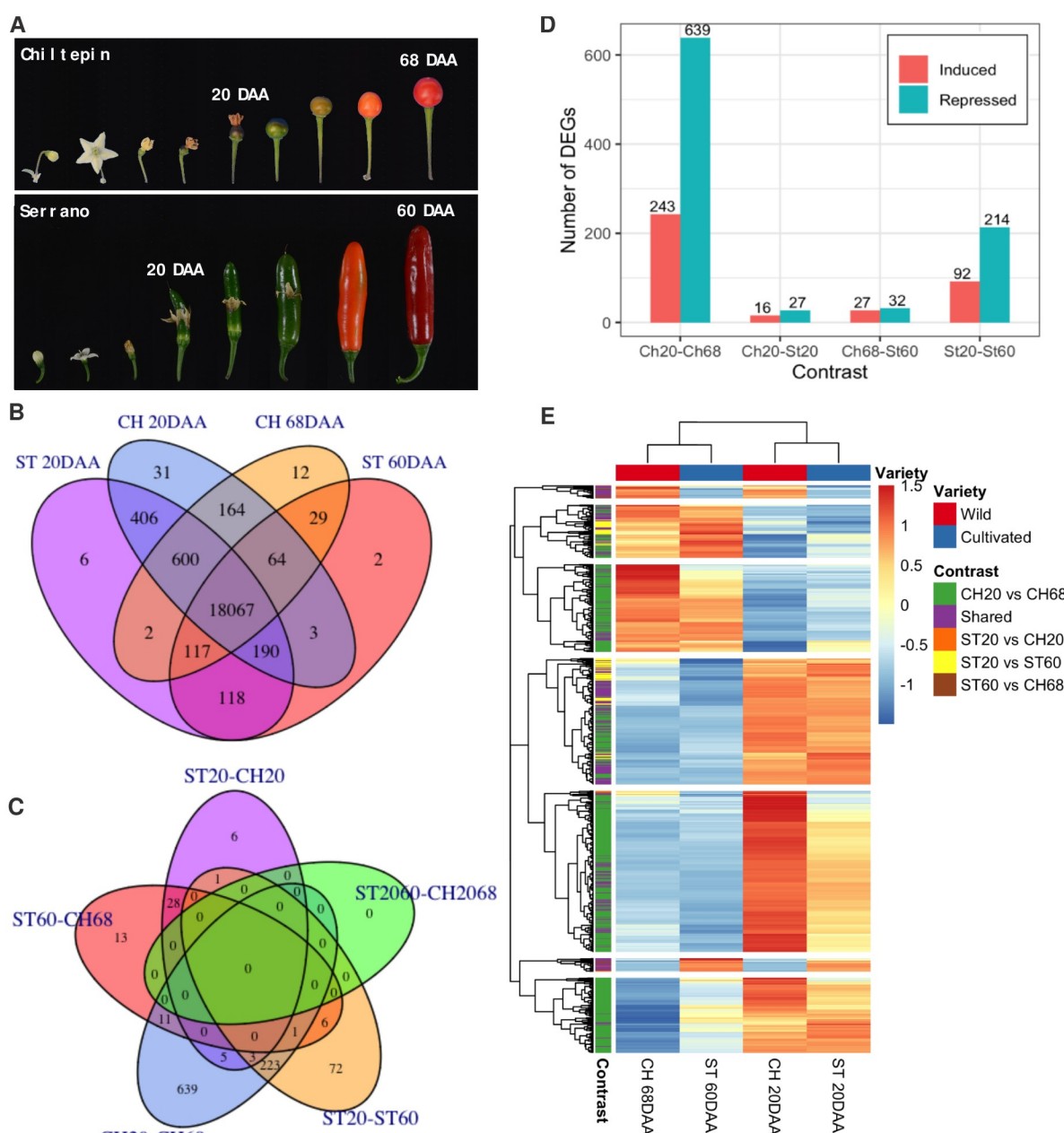

**Fig 1.** A) Chiltepin and Serrano fruits at different stages of development, marked with 20 DAA and 60/68 DAA the immature and mature stages sampled for the analysis performed in this study. B) Venn diagram for the number of genes expressed at each sampled stage of development for Chiltepin and Serrano. Numbers in each intersection represent the number of genes expressed in the sets corresponding to the intersection. C) Venn diagram for the number of differentially expressed genes (DEGs) per contrast; numbers in intersections represent the number of genes differentially expressed shared in the intersection contrasts. D) Number of downregulated and upregulated genes in each of the four contrasts. E) Heatmap representation of the DEGs found in at least one contrast, clustered by expression pattern of samples (columns) and genes (rows). Colored boxes at the beginning of every row indicate the contrast in which the corresponding gene is differentially expressed (DE); a purple box (shared) means the corresponding gene is DE in two or more contrasts. E) Heatmap representation of all DEGS clustered by expression pattern of samples (columns) and genes (rows); colored boxes indicated the contrast where the corresponding gene is differentially expressed.

expressed only at the mature stage, 2 genes were expressed only in ST and 12 in CH. The number of genes expressed is greater in both ST and CH at 20DAA.

## Differential expression

A differential expression analysis was performed on all contrasts (S2 Table). Comparing Chiltepin vs Serrano at the same development stage (Ch20-St20 and Ch68-St60) and mature vs green stages of both peppers (Ch20-Ch68 and St20-St60), we found 1,008 differentially expressed genes (DEGs) in at least one contrast. The number of DEGs per contrast ranged from 43 in the contrast Ch20-St20 to 882 in the contrast Ch20-Ch68. We found no DEGs shared by all contrasts; however, 278 genes were differentially expressed in more than one contrast and 730 genes were expressed only in one contrast. Most exclusive DEGs (639, 63.39%) were found in the contrast Ch20-Ch68; followed by 72 (7.14%) DEGs exclusive of the contrast St20-St60, 13 (1.29%) DEGs in the contrast Ch68-St60, and 6 DEGs (0.6%) exclusive of the contrast Ch20-St20 (Fig 1C and 1D). In a similar situation, we found that most DEGs in contrasts Ch20-Ch68 and St20-St60 were down-regulated (Fig 1D). In Ch20-Ch68 we found 639 (72.45%) down-regulated genes, and only 243 (27.55) DEGs were up-regulated. In contrast St20-St60 we found that 214 (69.94%) DEGs were down-regulated, while 92 (30.06%) DEGs were up-regulated. This is also the case of the contrast when we compared both varieties at the same development stage; in Ch20-St20, 27 (62.79%) and 16 (37.21%) of the DEGs were down and up-regulated, respectively; while in contrast Ch68-St60 32 (54.24%) and 27 (45.76%) DEGs were down-regulated and up-regulated, respectively (Fig 1D).

## Functional annotation and enrichment analysis

**Gene annotation.**   26,511 (73.88%) genes were annotated using blast2go (S3 Fig). E-value distribution showed that 75% of the annotated genes had alignments with E-values ≤ 6E-33. Six species contributed with 83.9% of the best hits used in the annotation: Arabidopsis thaliana alone contributed with 18,416 (58.15%) of the alignments, followed by *Oryza sativa subsp. japonica, Nicotiana tabacum, Solanum lycopersicum, Solanum demissum* and *Solanum tuberosum* with 1,248, 728, 578, and 441, respectively.

Tomato and chili pepper are close relatives; however, the size of the pepper genome is three times larger than the tomato genome. We expected to find multiple ortholog copies in the pepper genome, as it has been shown in previous studies [46]. For this, we used Inparanoid, a software capable of finding ortholog groups, where we identified 18,797 tomato orthologs. We searched for orthologs with another pepper genome (cv Zunla-1) with record in the Kyoto Encyclopedia of Genes and Genomes (KEGG) database. Further, 35,093 orthologs with a bitscore ≥ 50 were found using blastp.

**Gene ontology enrichment analysis.**   A total of 235,293 biological process, 104,106 cellular component, and 87,767 molecular function terms were assigned to the genes in our dataset based on the gene ontology (GO) slim database. Fig 2 shows the results of the GO enrichment analysis carried out with goseq. Two contrasts (Ch20-Ch68 and St20-St60) showed enriched GO terms. Most enriched terms correspond to the biological process ontology (17 and 7, respectively), followed by cellular component (4 and 2, respectively) and molecular function (2 and 0, respectively). The best-represented ontologies in both contrasts correspond to response to stress (420 and 159 DEGs), response to chemicals (421 and 157 DEGs), response to endogenous stimulus (313 and 114 DEGs), response to external stimulus (292 and 116 DEGs), and response to biotic stimulus (257 and 101 DEGs). The only two enriched GO terms in the cellular component in St20-St60 are also enriched in Ch20-Ch68 (Vacuole, 148 and 61 DEGs and cell wall, 117 and 41 DEGs). St20-St60 shares all its enriched GO terms with Ch20-Ch68;

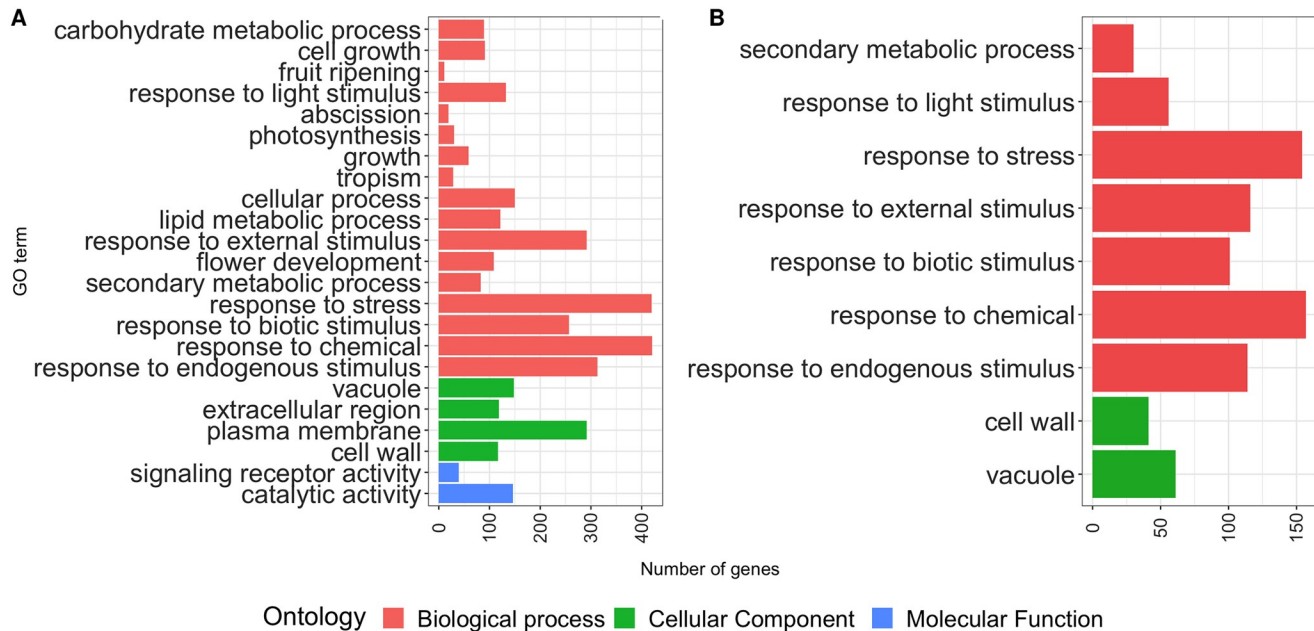

**Fig 2. Gene ontology enrichment analysis of differentially expressed genes (DEGs).** X-axis indicates the number of DEGs in the corresponding GO term, and the color of the bars represents the ontology of the corresponding GO term. A) Enriched GO terms in contrast Ch20-Ch68 (Chiltepin 20 vs 68 DAA), B) Enriched GO terms in contrast St20-St60 (Serrano 20 vs 60 DAA).

however, the Ch20-Ch68 possesses exclusive enriched terms. There are 10 enriched terms in the biological process ontology: cellular process (150 DEGs), lipid metabolic process (121 DEGs), flower development (108 DEGs), cell growth (91 DEGs), carbohydrate metabolic process (89 DEGs), growth (59 DEGs), photosynthesis (30 DEGs), tropism (28 DEGs), abscission (19 DEGs) and fruit ripening (11 DEGs). Likewise, there are terms enriched in the cellular component ontology: plasma membrane (292 DEGs), extracellular region (119 DEGs), and in the molecular function ontology: catalytic activity (146 DEGs) and signal receptor activity (39 DEGs).

**KEGG metabolic pathways enrichment analysis.** KEGG is a database of metabolic pathways maps used to obtain information of the metabolic networks corresponding to the genes in our dataset [47]. We used the *C. annuum* cv. Zunla-1 homologs to map the genes found to the KEGG metabolic pathway maps, and 135 pathways were found, including 9,401 enzymes out of 19,811 genes expressed (47.45%). Subsequently, we performed a metabolic pathway enrichment analysis with *ClusterProfiler*; an R package that implements a hypergeometric distribution to detect metabolic pathways enriched by the DEGs found in the present study against the universe of expressed genes. Fig 3 shows the enriched metabolic pathways detected, only the comparison between Ch20-Ch68 and St20-St60 displayed enriched pathways (EMP) statistically significant. *Phenylpropanoid biosynthesis*, *Photosynthesis-antenna proteins* and *Cutin, suberin, and wax biosynthesis* pathways were enriched in both contrasts and were also the most statistically significant; although those three EMPs showed enrichment in both contrasts. They displayed differences in expression in genes related to some enzymes involved in metabolic pathways.

*Phenylpropanoid biosynthesis* is the EMP with the greater count of DEGs in Ch20–68 (21) and the second one in St20-St60 (8). All DEGs (8) in St20-St60 are down-regulated; while in Ch20-Ch68 genes (8) related to enzymes *cinnamyl-alcohol dehydrogenase* and *caffeoyl shikimate esterase* are up-regulated (S4 Fig). The *cutin, suberin, and wax biosynthesis* showed

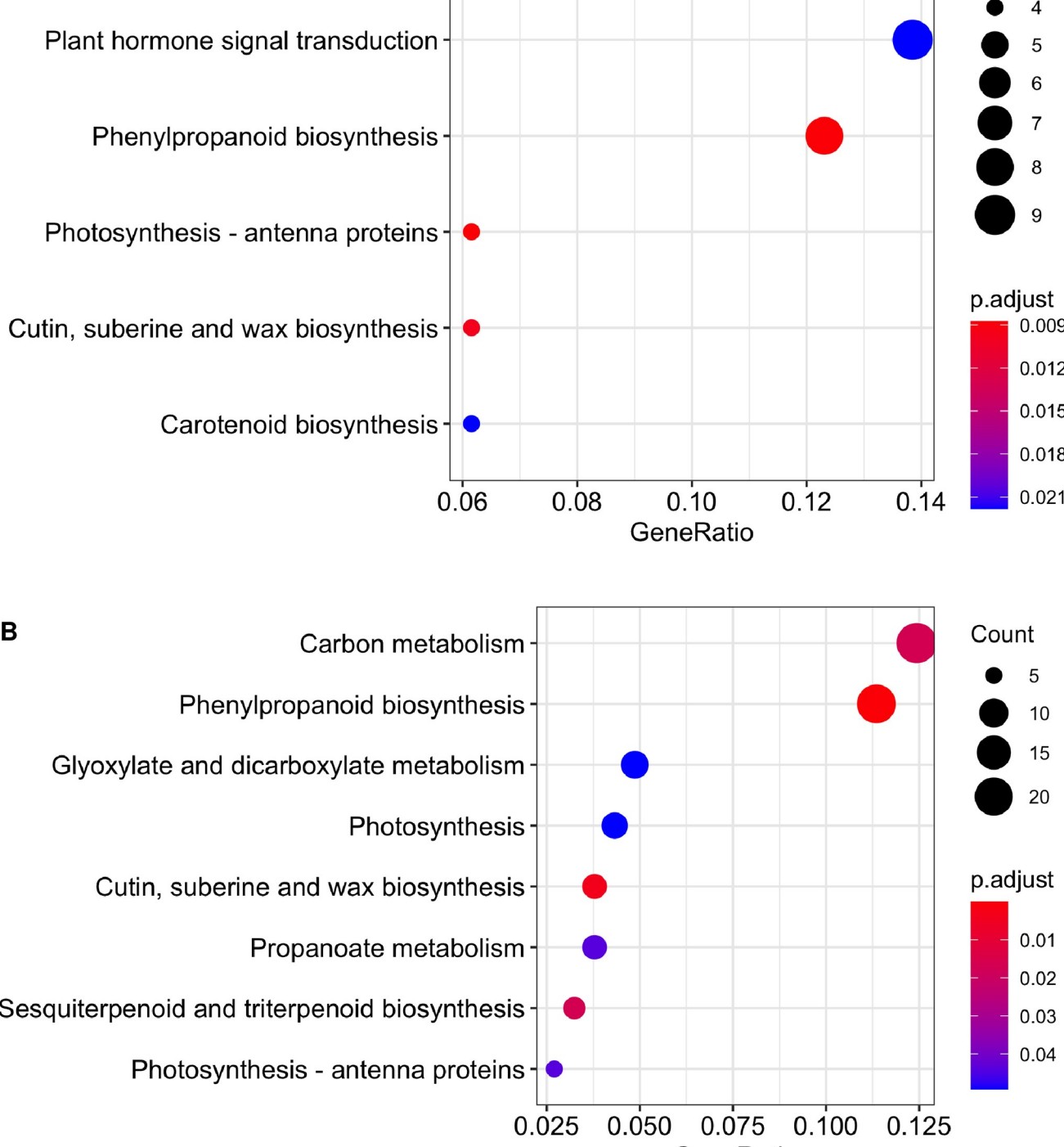

**Fig 3. Metabolic pathway enrichment of differentially expressed genes (DEGs).** X-axis indicates the gene ratio (the quotient of the number of DEGs and the number of genes in the background); the bubble size represents the number of DEGs in the corresponding pathway, and the color of the bubble indicates the enrichment q value of the corresponding pathway, lower q values mean more significant enrichment. A) enriched pathways of contrast St20-St60 (Serrano 20 vs 60 DAA). B) enriched pathways in contrast Ch20-Ch68 (Chiltepin 20 vs 68 DAA).

differences in genes corresponding to enzymes such as *fatty acid omega-hydroxy dehydrogenase* and *CYP94A5* upregulated only in Ch20-Ch68 and St20-St60, respectively. Gene *CYP77A6* was downregulated only in contrast Ch20-Ch68; all three genes are involved in the biosynthesis of unsaturated fatty acids, a crucial component of the cell wall polymers (S5 Fig). The last pathway enriched in both contrast is *Photosynthesis—antenna proteins* which showed differences only in the number of DEGs in each one; however, all genes are down-regulated in both contrasts, following the chlorophyll degradation that occurs over the fruit maturation process (S6 Fig).

EMPs exclusive to contrast St20-St60 include *Carotenoid biosynthesis* and *Plant hormone signal transduction*. *Carotenoid biosynthesis* contains 4 DEGs: *15-cis-phytoene synthase* (upregulated), *9cis-epoxycarotenoid dioxygenase* (upregulated) and *xanthoxin dehydrogenase* (downregulated) responsible for the production of phytoene, and *9-cis-epoxycarotenoid dioxygenase* (downregulated) responsible for the biosynthesis of Xanthoxin and Abscisic aldehyde (S7 Fig). EMP *Plant hormone signal transduction* displayed the greatest number of DEGs (9) in this contrast. Those included PP2C (upregulated) and SnRK2 (downregulated), part of ABA signaling, related to stomatal closure, seed dormancy, and carotenoid biosynthesis; ETRF1/2 (upregulated) part of the ethylene signaling a major player in fruit ripening and senescence. Finally, AUXIAA and SAUR were downregulated and upregulated, respectively; both are part of the auxin signaling pathway, related to cell enlargement and plant growth (S8 Fig).

EMPs found only in contrast Ch20-Ch68 include *Carbon metabolism* (23 DEGs), *Propanoate metabolism* (7 DEGs), *Glyoxylate and dicarboxylate metabolism* (9 DEGs), *Photosynthesis* (9 DEGs) and *Sesquiterpenoid and triterpenoid biosynthesis* with 6 DEGs; which coding for enzymes -germacrene D synthase (2 genes, downregulated and upregulated), *alpha-farnesene synthase* (upregulated), *squalene monooxygenase* (downregulated), *(3S,6E)-nerolidol synthase* (downregulated) and *beta-amyrin synthase* (downregulated), responsible for the production of defensive compounds germacrene D, $\alpha$ farnesene, terpenoid backbone, nerolidol and $\beta$ amyrin, respectively (S9 Fig).

**MapMan pathway analysis.** A MapMan analysis was performed on all the DEGs per contrast to identify groups of genes assigned to important biological processes. The most prevalent categories in contrasts Ch20-St68 and St20-St60 were RNA, signaling, protein, cell, development, transport, and stress. The most abundant categories in contrast Ch20-St20 were stress and protein. Categories with most DEGs in contrast Ch68-St60 were stress, signaling, RNA, and hormones. Fig 4 shows a summary of the most relevant MapMan pathways in each contrast. The MapMan pathways for contrast Ch20-St20 shows that most genes are downregulated, indicating that those genes present at least 4-fold times expression level in Chiltepin than in Serrano at 20 DAA. Most genes correspond to different responses to stress, including transcription factor WRKY44 related to the production of anthocyanin, catechin, and tannin; gene (E)-$\beta$-ocimene synthase a monoterpenoid produced as a response to herbivore attack [48].

The signaling category include protein COP9 involved in hormone signaling and development [49], GBP a GTPase involved in protection against pathogens by interferon responses [50], a BTB/POZ protein involved in regulating gene expression through the mediation of chromatin conformation [51], and an F-box protein; which acts as target detector for the ubiquitin-protein ligases SCF E3s, responsible for protein degradation, thus, allowing the cell to respond rapidly to changes in the environment [52]. Genes upregulated include the hsp70 gene, related to protein modification and protection from thermal or oxidative stress, and gene Ycf1, part of the intermediate translocation complex involved in protein import into the chloroplasts [53].

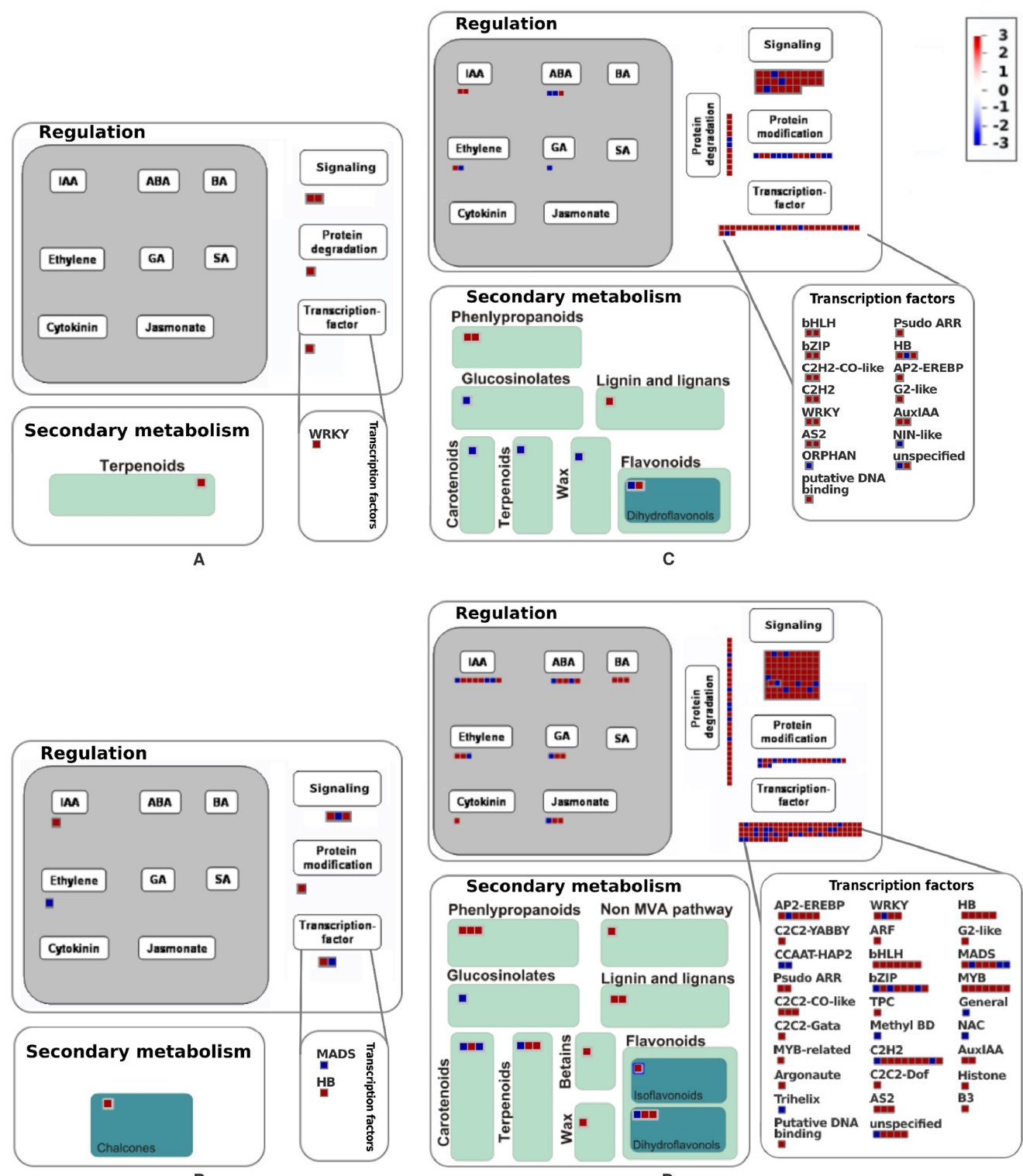

**Fig 4. MapMan summary of changes in the expression of differentially expressed genes (DEGs) in pathways regulation: Hormones, signaling, protein modification, protein degradation and transcription factors; and secondary metabolism: Terpenoids, phenylpropanoids, flavonoids, carotenoids, glucosinolates, etc.** Each square represents a separate DEG, red indicates upregulation and blue indicates downregulation of the respective gene in the corresponding contrast. A) MapMan summary of DEGs in contrast Ch20-St20 (Chiltepin vs Serrano at 20 DAA). B) MapMan summary of DEGs in contrast Ch68-St60 (Chiltepin vs Serrano at 68(60) DAA). C) Map Man summary of DEGs in contrast Ch20-Ch68 (Chiltepin 20 vs 68 DAA). D) MapMan summary of DEGs in contrast St20-St60 (Serrano 20 vs 60 DAA).

Fig 4B shows the MapMan summary for contrast Ch68-St60. Genes downregulated show a 4-fold expression level greater in Chiltepin than in Serrano at 68 (60) DAA; upregulated genes indicate a 4-fold expression level greater in Serrano than in Chiltepin at 68 (60) DAA. Genes with higher expression in Chiltepin include transcription factor BH involved in regulating ethylene, a Small Auxin Responsive Protein (SAUR32), related to apical hook development [54]; Chalcone isomerase (CHI), a key enzyme in the flavanones biosynthesis; and ILK1, a gene that acts as a signaling element in the pathogen resistance process through phosphorylation of MPK3 and MPK6 [55]. Genes upregulated include an ethylene receptor (ETR1), a key regulator of the ethylene signal transduction pathway [56]; SR1-P1, a Phototropic-responsive protein involved in plant defense by ubiquitination [57]; and the transcription factor FUL2, a MADS-Box protein regulator of several maturation processes like fruit softening, carotenoid accumulation, and cell wall modifications [58].

Fig 4C shows the summary of the MapMan pathways of contrast St20-St60. Transcription factors was the category with most DEGs (29). Only 4 genes were upregulated and 25 were downregulated. Some genes in this category are related to fruit development and ripening, DA1, an ubiquitin receptor that limits the final size of the seed by regulating cell proliferation [59]; 3 IAA TFs (IAA8, IAA27, IAA32, downregulated), involved in silencing of early auxin genes at low auxin concentration [60]; 2 WRKY TF (downregulated), related to stress response and production of tannins and anthocyanin [61]; 3 Homeobox TF (2 downregulated, 1 upregulated), related to negative regulation of cell elongation and proliferation [62]; and 2 bZIP TF (downregulated) that may be involved in growth regulation under water deficit [63]. The signaling category in contrast St20-St60 contained 23 DEGs. Downregulated genes (20) included gene LSH10, involved in cell growth promotion in response to light [64]; gene THE1, related to cell elongation during vegetative growth [65]; NPY2, essential in the auxin-mediated organogenesis [66]; gene TMK1, involved in AUX signal transduction and regulation of cell expansion and proliferation [67].

The hormone category contained 8 DEGs; 6 of them were downregulated and 2 upregulated. There were 3 ABA related TF, two of them (upregulated) are protein phosphatases 2C (PP2C) that may be involved in cell expansion via acid growth mechanism [68]. Also, 2 members of the PIN family (downregulated) auxin efflux facilitator, involved in auxin transport; 2 Ethylene related genes (ACCO2, downregulated and ERF1, upregulated), involved in the ethylene biosynthesis cascade; and a Gibberellin related protein SNAK1 (upregulated) with antimicrobial activity [69].

MapMan pathways for contrast Ch20-Ch68 is shown in Fig 4D. It contains 73 signaling related genes. Upregulated genes (7) include MAPKKK18, an ABA transducer in abiotic stress [70]. There were 66 downregulated genes, including ALE2, involved in differentiation of the protoderm into shoots epidermis and cuticle [71], GHR1 an ABA receptor required for regulation of stomatal movement [72], TMK1 and TMK3 involved in auxin signal transduction and regulation of cell expansion and proliferation [67]; gene LSH, which promotes cell growth in response to light [64]; gene NPY2, involved in auxin-mediated organogenesis [66]; THE1, a receptor required for cell expansion [65]; and 8 defense-related genes (SR1P1, M3K5G, NIK1, GLR34, NCL1, EDR2L, IQM3, PICBP).

Another important category in this contrast is plant hormone with 25 DEGs (17 downregulated, 8 upregulated). Here we found 3 LOX genes (LOX21 and LOX15 downregulated, LOX15 upregulated) involved in different processes of the plant like growth and development, pest resistance, and senescence [73]. We also found 2 ACCO genes (ACCO1 and ACCO2 both downregulated) related to ethylene biosynthesis, ACCO1 is a key gene in climacteric fruit ripening [74]; and 5 genes expressed as a response to stress (SNAK2, SC5D, and AHK4 downregulated, SNAK1 and PTI5 upregulated).

## Genes involved in fruit development and maturation

Chili pepper is a non-climacteric fruit, the molecular mechanisms of its fruit ripening process have not been completely explained; however, tomato, also part of the nightshade family and closely related to the *Capsicum* genus, is a climacteric fruit whose maturation process has been well studied using several ripening mutants; like non-ripening (nor), never-ripe (nr), colorless non-ripening (cnr), and ripening-inhibitors (rin). In this study, we selected 105 tomato genes involved in different parts of the fruit development and ripening process to evaluate their possible involvement in the maturation of chili pepper fruits. 778 candidate genes were found and 53 genes were differentially expressed in at least one contrast. Fig 5 shows a heatmap of the 53 DEGs. Clusters of genes and samples were calculated based on expression levels; samples of Chiltepín and Serrano were perfectly clustered together according to their developmental stage (20 DAA and 68,60 DAA), and 7 gene clusters were created.

Cluster 1 contains only an ERF6 gene, a transcription factor that negatively regulates ethylene and carotenoid biosynthesis [75] and is expressed at both stages in Serrano with a peak in 60DAA. Cluster 2 contains 13 DEGs with similar expression pattern: low expression at 20 DAA and high expression at 60(68) DAA. Although their similar expression pattern, most of these genes are DE only in contrast Ch20-Ch68; except for PSY1, a key enzyme in carotenoid biosynthesis [76] and PG2 that catalyzes the depolymerization of pectin in the pericarp [77] (upregulated in contrasts Ch20-Ch60 and St20-St60). This cluster also contains the transcription factor AGAMOUS-LIKE1 (TAGL1) regulator of carotenoid accumulation, pericarp thickness, and ethylene biosynthesis and one of the main regulators of genes PSY and PG [74].

The third cluster display a set of interesting genes, with expression in Serrano at both stages and only in 20DAA in Chiltepin. Genes APETALA2 (downregulated in Ch20-Ch68), an ethylene response element that modulates ripening involved in ethylene and carotenoid biosynthesis [78]. This cluster also contains two genes downregulated in Ch20-Ch68 and upregulated in Ch60-St68 (about the same expression at both stages in St). Gene FUL2 positive regulates several aspects of fruit maturation: fruit softening, carotenoid accumulation, sucrose metabolism, cell wall modification, cuticle formation, and ethylene biosynthesis [79]; and ACO3 one of the main enzymes involved in ethylene biosynthesis [80].

The fourth cluster contains only gene FSM1 expressed only in Ch at 20 DAA; this transcription factor seems to regulate several aspects of the young fruit growth (growth phase II) [81]. Cluster number 5 contains 4 DEGs downregulated in contrast Ch20-Ch68, including an OVATE, a gene important for being a negative regulator of fruit elongation responsible for round shape in wild tomatoes [82]; transcription factor MYB12, a regulator of enzymes *chalcone synthase* and *flavonol synthase*, important for the flavonoid biosynthesis [83]; another copy of gene TAGL1, and gene FAS, that controls locule number and regulates the expression of several genes; involved in processes like meristem and flower development, microtubule-binding activity and sterol biosynthesis [84].

Clusters 6 and 7 contain genes expressed at 20DAA but not at 60(68) DAA in both cultivars, all genes in cluster 6 are downregulated in Ch20-Ch68, and it includes ethylene, auxin, and Abscisic acid regulation related genes; cluster 7 contain 3 genes that were DE in contrast St20-St60 (downregulated), two copies of the auxin-responsive protein 9 (IAA9), a transcriptional factor that acts as a repressor of early auxin response genes [60], and the Inhibitor of Meristem Activity (IMA), a mini zinc finger protein that regulates the meristem activity and the flowering and ovule development process [85]. There are 10 genes DE in more than one contrast in cluster 7, all of them displayed similar expression patterns in all contrasts (both downregulated in Ch20-Ch68 and St20-St60).

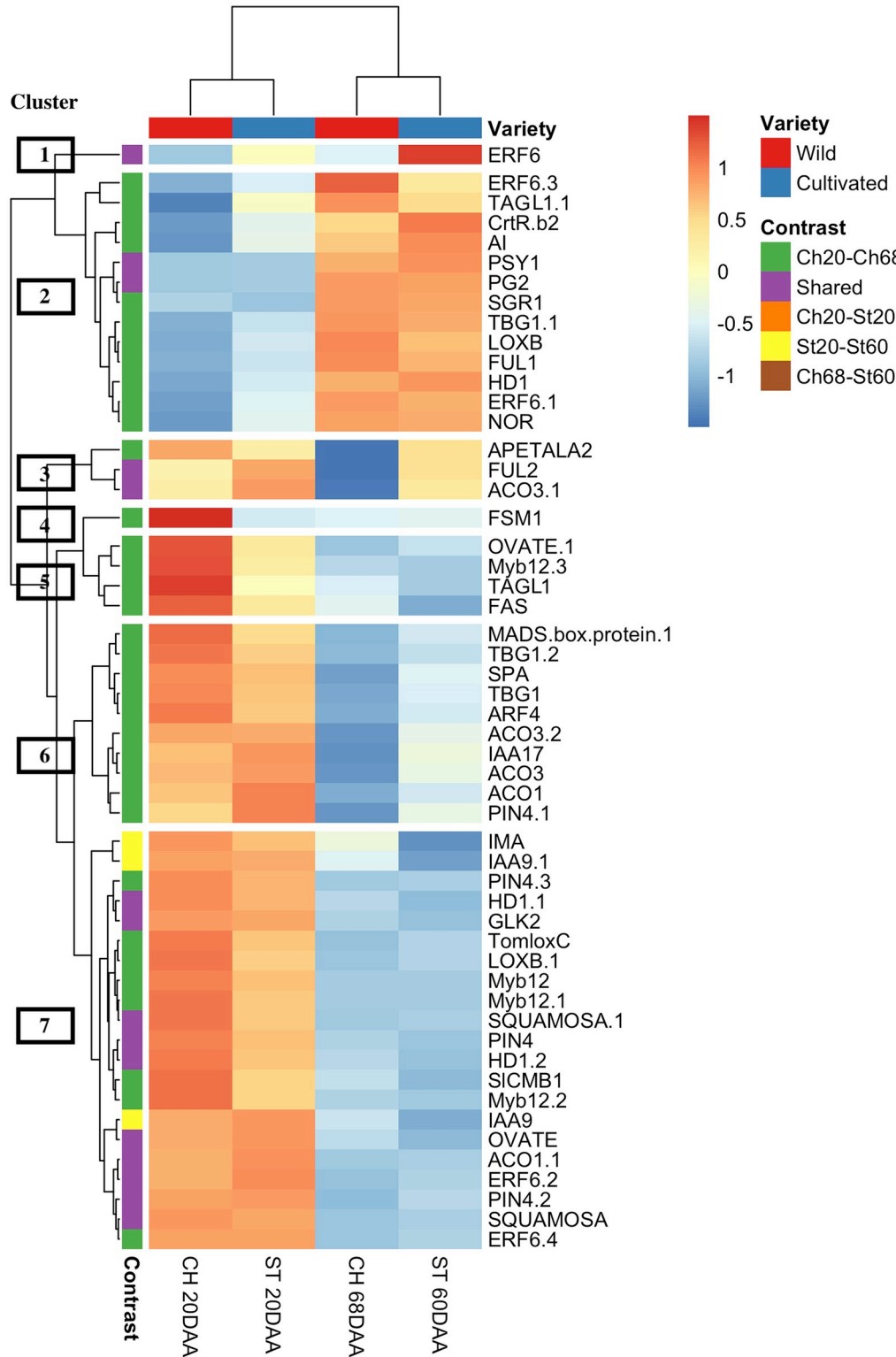

**Fig 5. Heatmap representation of the differentially expressed genes (DEGs) of pepper orthologs of tomato fruit development and ripening genes.** Expression pattern clustered by samples (columns) and genes (rows). Colored boxes at the beginning of every row indicates the contrast in which the corresponding gene is differentially expressed (DE), a purple box (shared) means the corresponding gene is DE in two or more contrasts.

## Discussion

The fruit undergoes several metabolic and biochemical changes that produce the organoleptic properties, some of them assist in seed dispersal and provide nutritious components essential for human diets [86]. Unlike its close relative tomato, which is a model for climacteric fleshy fruit development and ripening [87]; chili pepper is a non-climacteric fruit, and its ripening mechanisms have not been completely understood. Like in other non-climacteric fruits, ethylene has no effect triggering ripening; but in some peppers it seems to be involved in regulating several aspects of the ripening process. Previous studies have shown that non-climacteric ripening of chili peppers might be regulated by the phytohormones ethylene and abscisic acid (ABA) [88, 89].

In this study, we analyzed RNASeq data of a wild and a cultivated variety of chili pepper at an immature and full mature stage in order to identify differentially expressed genes and differences in gene expression patterns between the two cultivars. Our main findings suggest that while both cultivars display genes with very similar expression patterns, key differences in gene expression appear related to fruit development as we summarize below.

Both varieties displayed a greater number of expressed genes in the immature stage than in the mature stage, previous studies have shown that most of the differentially expressed genes over fruit development and ripening were down-regulated (expressed at fruit development and turned off at mature stage), suggesting this might be caused by the proximity of the senescence [44].

GO enrichment analysis showed no enriched terms in contrasts of the two peppers at the same time (Ch20-St20, Ch68-St60). The more enriched terms were related to response to stress/pathogens/light; which makes sense under the knowledge that fruit maturation is regulated by the same hormone cascades used as defense response and is quite complicated to separate both processes. KEGG analysis showed no metabolic pathways enriched for contrast Ch20-St20 and Ch68-St60; however, in contrast St20-St60 the MP *hormone and signal transduction* was enriched with DE genes regulators of the biosynthesis of ethylene (ERF1), abscisic acid (PP2C, SnRK2), and auxin (IAA, SAUR50). ABA related gene PP2C (upregulated) is a negative regulator of ABBA, and SnRK2 (downregulated) is a positive regulator of ABA in the 'PYR-PP2C-SnRK2' cascade signaling. Previous studies have shown that along with the ripening process, the expression levels of SnRK2 decreases while PP2C level increases [88], consistently with our results.

Commercially, fruit size is one of the most important traits in the pepper fruits. We found that the protein encoded by gene PP2C acts with gene SAUR50 (upregulated) from the auxin pathway; SAUR50 reduces the activity of D-clade type 2C protein phosphatases (PP2C-Ds), responsible for phosphorylation of the second amino acid in the plasma membrane H +-ATPases (PM H+-ATPases). Reduction in the activity of PP2C-Ds proteins activates cell elongation by H+-ATPases via an acid growth mechanism [54, 68, 90]. This has been tested in various type of organs but not in fruits and could be responsible for the augmented size of Serrano fruits. Technically, this set of genes may work simultaneously since the cell expansion stage, but it is impossible to know since our sample points are at the fringes of that stage. SAUR50 is regulated by gene FRUITFULL (FUL2), expressed in St60 but not in Ch68. Previous studies have shown that FUL2 by itself is related to an increase in fruit size [58], reduces post-harvest, slowest dehydration, and regulates ethylene biosynthesis key genes ACO1 and ACS2 [91]. We found that a copy of ACO3 displayed the same expression pattern as FUL2 and is included in the same cluster.

Fruit shape is another trait with obvious differences between the two analyzed peppers. Chiltepin in the Sonoran region presents a small and rounded shape, while Serrano

'Tampiqueño 74' displays a big and elongated shape. We found an ortholog of OVATE in our dataset expressed in Ch20 DAA and with lower expression levels (under 4 FC) in St20 DAA. In pepper, an OVATE-like protein has been associated with changes in fruit shape [92]. OVATE is a negative regulator of cell growth via suppression of GA biosynthesis, and its over-expression leads to dwarf fruits [93]; downregulation of OVATE expression in pepper turned round fruits into oblong forms [92]; OVATE was clustered together with a homolog of the tomato FAS gene, a CLAVATA3 (CLV3) homolog that in tomato contains a mutation that produces bigger fruits with a fasciated shape; however, the original function of CLV3 was to restrict the size of the stem cell population. In Chiltepin, OVATE and FAS/CLV3 could be responsible for the round shape and small size of the Chiltepin fruit.

Secondary metabolites are compounds that are not necessary for the basal processes in plants but are useful for interaction with the surrounding environment. Usually, secondary metabolites are used as a defense against different types of stress. There is a great range of secondary metabolites synthesized from primary metabolites (amino acids, lipids, and carbohydrates) including terpenes, phenolics, nitrogen (N), and sulfur (S) containing compounds. Additionally, secondary metabolites contribute to the organoleptic characteristics of pepper fruit (odor, taste, color, and pungency), and are an excellent source of nutritious elements essential for human diets [94, 95]. In our study, we found differences in gene expression patterns between the two peppers; usually, most genes involved in secondary metabolism are turned on at developmental stages, and start shutting off over the ripening stage; once the pigments, volatile compounds, capsaicinoids, etc, have been stored and the fruit has obtained the traits that make it attractive for seed dispersion. We found a very similar expression pattern in both peppers; however, two genes were DE only in Chiltepin: Chalcone isomerase (CHI3) involved in the anthocyanin biosynthesis (downregulated in contrast Ch20-St20 and Ch20-Ch68); (E)-$\beta$-ocimene synthase, a monoterpenoid that has been associated to herbivore attack response [48] (downregulated in contrast Ch68-St60); and $\beta$-carotene hydroxylase-2 (BCH2), an enzyme responsible for the xanthophyll biosynthesis [96] (upregulated in contrast Ch20-Ch68). This may suggests there could be a difference in secondary metabolites accumulation. Previous studies have shown that upregulation of Chalcone isomerase in tomato produced an increase of up to 78-fold in fruit flavonols [97].

Fig 6A and 6B show a hypothetical biological model for the fruit maturation process of Serrano and Chiltepin pepper, respectively, based on the results obtained in this study. There were notable differences in both peppers related to fruit development and growth, secondary metabolism, cell wall modification response to stress and seed development. In Serrano (Fig 6A) pepper, induced cell growth genes (*FUL2,SAUR50 AND PP2C*) may be responsible for the phenotypical elongated shape via an acid growth mechanism, those genes were not induced in Chiltepin (Fig 6B) in which a combination of two genes (*OVATE* and *FAS/CLV3*) might be responsible for the small and round fruits. In Serrano most genes related to secondary metabolism are repressed, probably because this pepper presents an early senescence. On the other hand, in Chiltepin there were induced genes related to the biosynthesis of carotenoids, terpenoids, flavanones, and phenylpropanoids. There were induced genes in Serrano related to seed development that were not differentially expressed in Chiltepin. Finally, both peppers presented a great number of genes related to stress responses; however, those were more diverse in Chiltepin than in Serrano, and in some cases there were more than one differentially expressed gene with the same annotation (probably paralog genes), this was also the case for genes responsible for cell wall degradation in Chiltepin, where more than one gene were found differentially expressed.

In conclusion, we presented a comprehensive transcriptome analysis of a wild and a cultivated pepper variety over two developmental stages. Our results showed that despite the great

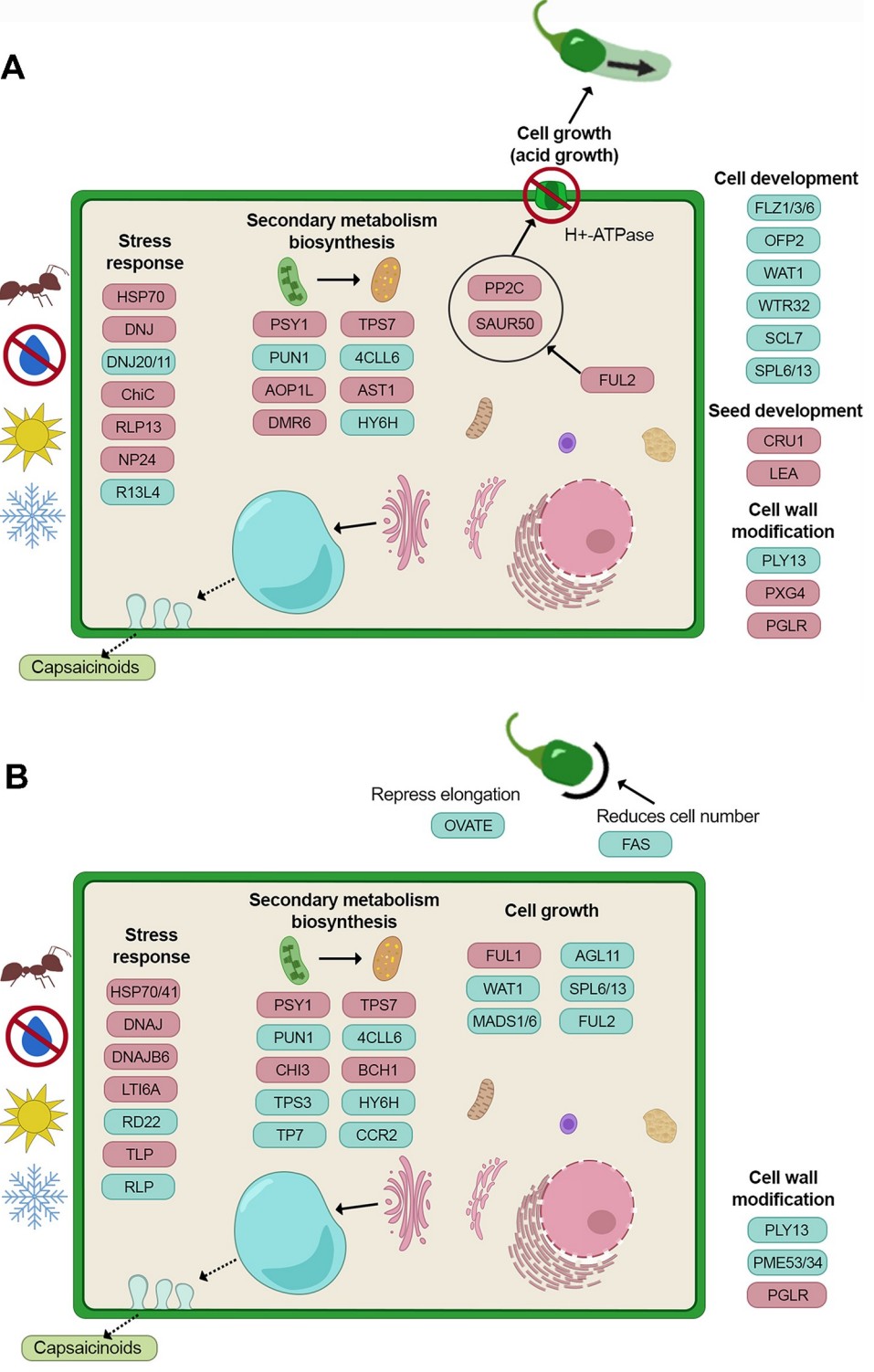

**Fig 6. Biological model for the fruit maturation process of A) Serrano pepper (St20-St60), and B) Chiltepin pepper (Ch20-Ch68).** Red boxes indicate genes induced (with at least 4-fold expression at the mature stage), blue boxes indicate gene repressed (with at least 4-fold expression at the immature stage).

morphological differences between the fruits of these two varieties, they showed very similar expression patterns, with differences in genes related to shape, size, ethylene and secondary metabolites biosynthesis; suggesting that changes produced by the domestication of chili pepper may be very specific to the expression pattern of genes involved in traits important for commercial fruits.

## Materials and methods

### Plant material

Fruits from the wild pepper *Capsicum annuum var. glabriusculum* 'chiltepin' were collected from a wild population in Tepoca locality, Municipality of Yécora (Sonora, Mexico; 28.43˚ N, -109.25˚ W). Seeds were germinated in plastic pots containing sterile soil mixture as previously described [98]. The seedlings were grown under greenhouse conditions in a randomized experimental condition at the Laboratorio Nacional de Genómica para la Biodiversidad (LANGEBIO-CINVESTAV, Irapuato-México). Individual flowers were tagged at anthesis, fruits were randomly collected from different plants at 20 (green fruits, immature state) and 68 (red fruits, mature state) DAA. After sampling, the fruits were cleaned with ethanol and stored at -80˚C until further analysis.

### RNA extraction, library construction and sequencing

For total RNA, 6 fruits at 20 and 68 DAA stages each were randomly selected from the pool of all harvested fruits. Whole fruits (pericarp, placenta, and seeds) were ground using a TissueLyser system (QIAGEN) with beads. Disruptions were performed at intervals of 1 min at 30 freq/s. RNA extraction was performed from each fruit of Chiltepin separately, and 100mg aliquots were measured from each RNA extraction and then pooled. The process was repeated with a different collection of fruits to get an independent biological replicate for a total of 4 samples. A NucleoSpin RNA Plant kit (Macherey-Nagel) was used for total RNA extraction following the manufacturer's recommendations. Each RNA sample was subjected to DNase treatment using DNase I (Macherey-Nagel) according to the manufacturer's protocol. Total RNA concentration was quantified using a Nanodrop ND-1000 spectrophotometer (Thermo Scientific Nanodrop). RNA integrity was evaluated by electrophoresis on a 1.0% agarose gel, and aliquots of RNA were run in an Agilent 2100 Bioanalyzer (Agilent, Santa Clara, CA, USA).

For sequencing, cDNA libraries were created using 12 *μg* of total RNA from each biological replicate. Sequencing of the four total RNA samples was performed in the Laboratorio de Servicios Genómicos (CINVESTAV-LANGEBIO, Irapuato, Mexico) using Illumina TruSeq RNA sample v2 Guide preparation, following the manufacturer's instructions. The four cDNA libraries were sequenced from both 5' and 3' ends using an Illumina NextSeq 500 platform, according to the manufacturer's instructions; to produce 260 million 2x150-bp paired-end reads (See S1 Table, S1 Fig for details). Quality filtering, alignment, and abundance calculation.

### Quality filtering, alignment and abundance calculation

Raw sequencing reads were filtered using Trimmomatic V. 0.3678 [99]. Illumina sequencing adapters were removed, and a sliding window of 4 base pairs and a Phred quality value < 15 was used to trim reads. Finally, reads with a final length < 50 base pairs were removed (S1 Table). Raw and trimmed reads were subjected of a quality control analysis using the FastQC V. 0.11.5 software (https://www.bioinformatics.babraham.ac.uk/projects/fastqc/; last accessed

June 2020) to verify the per-base sequence quality, length distribution, per sequence GC content, and the absence of adapter (S1 and S2 Figs).

Trimmed reads were aligned to the chili pepper cv. CM334 reference genome (version 1.6) [100] using the alignment software Hisat2 V. 2.1.0 [101] with known splice sites extracted from the *Capsicum annuum* cv. CM334 annotation (version 2.0, http://peppergenome.snu.ac. kr; last accessed June 2020) and default parameters (S1 Table). Gene raw counts were estimated using HTseq-count V. 0.11.0 [102] with the 'union' option.

## Sequence annotation

Annotation of the *Capsicum annuum* cv. CM334 genes was accomplished using the software Blast2GO [103] by performing a BLAST search against the reviewed database UniProtKB/ Swiss-Prot [104] to identify similar sequences. GO and the generalized slimGO terms were assigned by Blast2GO with default parameters.

The *Capsicum annuum* genome has been through unequal gene-duplication events, resulting in a genome four times bigger than the tomato [100]. A clustering ortholog approach was used to infer orthology relationships between pepper and tomato genes; the software Inparanoid V. 4.2 [105] was used with the tomato reference genome protein sequence (ITAG Release 3.2).

Metabolic pathway assignments based on the Kyoto Encyclopedia of Genes and Genomes (KEGG) database (http://www.genome.jp/kegg/, last accessed June 2020) [106] were set using the KAAS automatic annotation server [47], and by mapping against the *Capsicum annuum* cv. Zunla-1 genome [107] using the bi-directional best hit method.

## Statistical analyses

The Bioconductor package edgeR v. 3.24.3 [108] in R v. 3.5.3 [109] was used to evaluate the differential expression on contrasts between accessions: Chiltepin (CH, Wild) and Serrano (ST, Cultivated), and sampling time: 20, 60 and 68 DAA (S2 Table). Genes with zero total count and genes with average log count per million < 1 in all libraries were excluded from the analysis. Normalization by trimmed mean of M values (TMM) was used to calculate normalization factors for each library. We estimated common, trended and tagwise dispersion, and a generalized linear model (GLM) with a negative binomial distribution was fit to raw count data. Genes with a false discovery rate (FDR) < 0.01 were classified as significantly differentially expressed (DE).

Gene ontology (GO) enrichment analysis was performed on DE genes from each contrast using the Bioconductor library goseq v. 1.38.0 [110] to test for under and over-representation of GO terms among the set of DE genes. Gene ontologies significantly enriched (with FDR < 0.05, ancestral terms included) were selected.

Metabolic pathway analysis was performed to assess possible biological processes that may be affected by the conditions (accession and sampling time) of the experiment. A list of enriched KEGG metabolic pathways (FDR < 0.05) was generated for each contrast using the R library ClusterprofileR v. 3.10.1 [111].

## Serrano 'Tampiqueño 74' dataset

A table of raw counts of Serrano 'Tampiqueño 74' samples was obtained from an ongoing research project [45]. The protocol described for Chiltepin was used; however, Tophat2 was used to align the clean reads against the reference genome instead of Hisat2.

## Supporting information

**S1 Fig. Per base quality of raw Chiltepin RNA-Seq reads.** Per base quality of raw Chiltepin RNA-Seq lectures: A) 20 DAA replicate 1, B) 20 DAA replicate 2, C) 68 DAA replicate 1, D) 68 DAA replicate 2. In all cases the plot on the left shows the FastQC result for forward raw reads and the plot on the right shows the FastQC results for reverse raw sequencing reads.
(PDF)

**S2 Fig. Per base quality of filtered Chiltepin RNA-Seq reads.** Per base quality of filtered Chiltepin RNA-Seq lectures: A) 20 DAA replicate 1, B) 20 DAA replicate 2, C) 68 DAA replicate 1, D) 68 DAA replicate 2. In all cases the plot on the left shows the FastQC result for forward trimmed reads and the plot on the right shows the FastQC results for reverse trimmered sequencing reads.
(PDF)

**S3 Fig. *Capsicum annuum* L. cv. 'Criollo de Morelos' gene annotation statistics.** Statistics of *Capsicum annuum* L. cv. 'Criollo de Morelos' genes annotated via BLAST2GO. A) BLAST2GO gene tag distribution, displays the number of genes with no Blast, blasted, annotated, and mapped. B) BLAST2GO GO level distribution, shows the number of annotations per GO level for biological process, molecular function and cellular component ontologies.
(PDF)

**S4 Fig. Enriched metabolic pathway *Phenilpropanoid biosynthesis*.** Metabolic pathway enriched in the contrast Ch20-Ch68, green boxes indicate enzymes encoded by genes repressed, red boxes indicate enzymes encoded by genes induced in the contrast.
(PDF)

**S5 Fig. Enriched metabolic pathway *cutine, suberine and wax biosynthesis*.** Metabolic pathway enriched in the contrast: A) St20-St60 (Serrano 20 DAA vs Serrano 60 DAA), B) Ch20-Ch68 (Chiltepin 20 DAA vs Chiltepin 68 DAA). Green boxes indicate enzymes encoded by genes repressed in the corresponding contrast, red boxes indicate enzymes encoded by genes induced in the corresponding contrast.
(PDF)

**S6 Fig. Enriched metabolic pathway *Photosynthesis-antenna proteins*.** Metabolic pathway enriched in the contrast: A) St20-St60 (Serrano 20 DAA vs Serrano 60 DAA), B) Ch20-Ch68 (Chiltepin 20 DAA vs Chiltepin 68 DAA). Green boxes indicate enzymes encoded by genes repressed in the corresponding contrast, red boxes indicate enzymes encoded by genes induced in the corresponding contrast.
(PDF)

**S7 Fig. Enriched metabolic pathway *Carotenoid biosynthesis*.** Metabolic pathway enriched in the contrast St20-St60 (Serrano 20 DAA vs Serrano 60 DAA). Green boxes indicate enzymes encoded by genes repressed in the corresponding contrast, red boxes indicate enzymes encoded by genes induced in the corresponding contrast.
(PDF)

**S8 Fig. Enriched metabolic pathway *Plant hormone signal transduction*.** Metabolic pathway enriched in the contrast St20-St60 (Serrano 20 DAA vs Serrano 60 DAA). Green boxes indicate enzymes encoded by genes repressed in the corresponding contrast, red boxes indicate enzymes encoded by genes induced in the corresponding contrast.
(PDF)

**S9 Fig. Enriched metabolic pathway *Sesquiterpenoid and triterpenoid biosynthesis*.** Metabolic pathway enriched in the contrast Ch20-Ch68 (Chiltepin 20 DAA vs Chiltepin 68 DAA). Green boxes indicate enzymes encoded by genes repressed in the corresponding contrast, red boxes indicate enzymes encoded by genes induced in the corresponding contrast.
(PDF)

**S1 Table. Chiltepin RNA-Seq reads statistics.** Number of Chiltepin RNA-Seq raw sequencing reads, number of filtered reads and alignment rate of Chiltepin trimmed reads against the reference genome *Capsicum annuum* (cv. 'Criollo de Morelos 334') per sample.
(XLSX)

**S2 Table. Contrast design matrix.** Contrast designs used in the differential expression analysis in EdgeR, contrasts with no biological meaning were dismissed.
(XLSX)

**S3 Table. Homologs with tomato fruit maturation genes.** Tomato genes related to fruit development and ripening and pepper homologs detected with BLASTp, E-value and Bitscore of each alignment.
(XLSX)

## Acknowledgments

The authors thank Juan Bautista Teran Fraijo, María del Socorro Fraijo Encinas, Oscar Julio Luna, Cecilia Piri Alcaraz, Lucia Preciado Gamez and Hugo Piri for administrative support during the project.

## Author Contributions

**Conceptualization:** Corina Hayano-Kanashiro, Octavio Martínez.

**Data curation:** Fernando G. Razo-Mendivil.

**Formal analysis:** Fernando G. Razo-Mendivil, Corina Hayano-Kanashiro, Octavio Martínez.

**Funding acquisition:** Corina Hayano-Kanashiro.

**Investigation:** Fernando G. Razo-Mendivil, Fernando Hernandez-Godínez, Corina Hayano-Kanashiro, Octavio Martínez.

**Methodology:** Fernando G. Razo-Mendivil, Fernando Hernandez-Godínez, Corina Hayano-Kanashiro, Octavio Martínez.

**Project administration:** Corina Hayano-Kanashiro.

**Resources:** Corina Hayano-Kanashiro.

**Software:** Fernando G. Razo-Mendivil.

**Supervision:** Corina Hayano-Kanashiro, Octavio Martínez.

**Validation:** Fernando G. Razo-Mendivil, Corina Hayano-Kanashiro.

**Visualization:** Fernando G. Razo-Mendivil, Corina Hayano-Kanashiro.

**Writing – original draft:** Fernando G. Razo-Mendivil, Corina Hayano-Kanashiro.

**Writing – review & editing:** Fernando G. Razo-Mendivil, Corina Hayano-Kanashiro, Octavio Martínez.

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
