## [Decision Letter · Decision Letter 0]

10 Jun 2021

PONE-D-21-14082

Transcriptomic analysis of a wild and a cultivated varieties of Capsicum annuum over fruit development and ripening

PLOS ONE

Dear Dr. Martínez,

Thank you for submitting your manuscript to PLOS ONE. After careful consideration, we feel that it has merit but does not fully meet PLOS ONE’s publication criteria as it currently stands. Therefore, we invite you to submit a revised version of the manuscript that addresses the points raised during the review process.

Both reviewers have suggested minor revisions in certain part of the manuscript.  Please respond to their comments.  

We look forward to receiving your revised manuscript.

Kind regards,

Xiang Jia Min, Ph. D.

Academic Editor

PLOS ONE

Journal Requirements:

1. Please ensure that your manuscript meets PLOS ONE's style requirements, including those for file naming. The PLOS ONE style templates can be found at https://journals.plos.org/plosone/s/file?id=wjVg/PLOSOne_formatting_sample_main_body.pdf and  https://journals.plos.org/plosone/s/file?id=ba62/PL.

“The authors thank Juan Bautista Teran Fraijo and Mar´ıa del Socorro Fraijo Encinas for  helping us secure financial support from COFUPRO. We also thank Oscar Julio Luna,  Cecilia Piri Alcaraz, Lucia Preciado Gamez and Hugo Piri for administrative support during the project. FR-M acknowledges the Mexican Council of Science and Technology CONACyT) for supporting with a PhD scholarship (261122) during the development of the project.”

“FR-M acknowledges the Mexican Council of Science and Technology (CONACyT) for supporting with a PhD scholarship (261122) during the development of the project.”

4. We note that Figure(s) S6 in your submission contain copyrighted images. All PLOS content is published under the Creative Commons Attribution License (CC BY 4.0), which means that the manuscript, images, and Supporting Information files will be freely available online, and any third party is permitted to access, download, copy, distribute, and use these materials in any way, even commercially, with proper attribution. For more information, see our copyright guidelines: http://journals.plos.org/plosone/s/licenses-and-copyright.

   a) You may seek permission from the original copyright holder of Figure(s) [#] to publish the content specifically under the CC BY 4.0 license.

“I request permission for the open-access journal PLOS ONE to publish XXX under the Creative Commons Attribution License (CCAL) CC BY 4.0 (http://creativecommons.org/licenses/by/4.0/) . Please be aware that this license allows unrestricted use and distribution, even commercially, by third parties. Please reply and provide explicit written permission to publish XXX under a CC BY license and complete the attached form.”

Reviewers' comments:

Reviewer's Responses to Questions

**Comments to the Author**

1. Is the manuscript technically sound, and do the data support the conclusions?

Reviewer #1: Yes

Reviewer #2: Yes

2. Has the statistical analysis been performed appropriately and rigorously? 

Reviewer #1: Yes

Reviewer #2: Yes

3. Have the authors made all data underlying the findings in their manuscript fully available?

Reviewer #1: Yes

Reviewer #2: Yes

4. Is the manuscript presented in an intelligible fashion and written in standard English?

Reviewer #1: Yes

Reviewer #2: Yes

5. Review Comments to the Author

Reviewer #1: The study addresses an important question of identifying genes with potential for improving commercially valuable traits including disease/drought resistance in an economically important vegetable crop (Capsicum annuum).

Line 7 I was confused by the sentence. Add “which consists of “ after domestication syndrome in the sentence: “Domesticated plants display a domestication syndrome, a significant increase in the size of their harvest parts, loss of dispersal ability, seed dormancy, and protection against herbivory” so that it is clear those attributes refer to the domestication syndrome if that is the case.

Lines 104-105 paragraph belongs to Discussion “Previous studies have reported similar results and suggest that genes are shutting off as the fruit reaches full maturity and senescence [44].”

Lines 117-120 paragraph belongs to Discussion

“Previous studies have shown that most of the differentially expressed genes over fruit 117 development and ripening were down-regulated (expressed at fruit development and 118 turned off at mature stage). Suggesting this might be caused by the proximity of the 119 senescence [44]. “

Line 119 need to remove the period . add comma , before “Suggesting this might be caused by the proximity of the…”

Lines 291-295 paragraph belongs to Discussion

“Chili pepper is a non-climacteric fruit, the molecular mechanisms of its fruit ripening 291 process have not been completely explained; however, tomato, also part of the 292 nightshade family and closely related to the Capsicum genus, is a climacteric fruit 293 whose maturation process has been well studied using several ripening mutants; like 294 non-ripening (nor), never-ripe (nr), colorless non-ripening (cnr), and ripening-inhibitors 295 (rin).”

Lines 353-356 Paragraph needs to be rewritten for ex. Our main findings suggest that while both cultivars display genes with very similar expression patterns, key differences in expression in genes appear related to fruit development as we summarize below

Reviewer #2: The manuscript by Razo-Mendivil et al. describes a genome wide transcriptome analysis of a comprehensive transcriptome analysis of a wild and a cultivated pepper variety over two developmental stages. Using short reads sequencing technologies, the authors identified the expressed pepper genes. The two peppers showed very similar gene expression patterns; differences in expression patterns of genes related to shape, size, ethylene and secondary metabolites biosynthesis.

Although several minor revisions have to be incorporated in the manuscript most of it is clearly written. The discussion and conclusions are supported by the data and the abstract and title reflects what has been found.

Minor revision:

In the Results section, Transcriptomic analysis of the wild Chiltepin (CH; Capsicum annuum var. glabriusculum) and the cultivated Serrano Tampique˜no (ST; Capsicum annuum L. cv. ‘Tampique˜no 74’) was performed at two stages (Fig 1A), the immature stage at 20 days after anthesis (DAA) for both CH and ST, and at mature state at 60 DAA for ST and 68 DAA for CH [45]. I did not find that the fruit of ST in Fig. 1A. Moreover, Please check if the reference [45] is appropriate.

6. PLOS authors have the option to publish the peer review history of their article (what does this mean?). If published, this will include your full peer review and any attached files.

Reviewer #1: No

Reviewer #2: No

---

## [Author Response · Author response to Decision Letter 0]

18 Jul 2021

Our response to reviewer has been already uploaded as file "ResponseToReviewers.pdf". Here we reproduce that file without the PDF format:

Response to Reviewers

Manuscript “Transcriptomic analysis of a wild and a cultivated varieties of Capsicum annuum over fruit development and ripening” (PONE-D-21-14082).

Dear Dr. Xiang Jia Min,

We thank you and the two reviewers for the time and effort invested in the review of our manuscript. Below we address all the points raised. Those points were taken into account to write the new version of the manuscript

Editor requirements

1. We have ensured that the new version of the manuscript meets PLOS ONE’s style requirements.

2. We used PLOS ONE’s LaTeX template in the new version of the manuscript.

3. Funding-related text was removed from the Acknowledgments section. Funding Statement: This

work was supported in part by COFUPRO (A/GTO/RGAG-2014-076-Consorcio de Fundaciones PRODUCE). FR-M acknowledges the Mexican Council of Science and Technology (CONACyT) for supporting with a PhD scholarship (261122) during the development of the project. The funders had no role in study design, data collection and analysis, decision to publish, or preparation of the manuscript.

4. Copyrighted images. Figures S4, S5, S6, S7, S8 and S9 contain copyrighted material (KEGG pathway map images). Table below gives details of those figures.

[TABLE NOT SHOWN HERE] 

As instructed by the Editor, we obtained written permission from the copyright holder to publish these figures (file “KEGG Copyright Permission 210739.pdf”).

Reviewer #1

1. “Line 7 I was confused by the sentence”. The line was modified by adding “which consists of ”.

2. “Lines 104-105 paragraph belongs to Discussion.” Those lines were moved to Discussion.

3. “Lines 117-120 paragraph belongs to Discussion.” Those lines were moved to Discussion.

4. “Line 119 need to remove the period.” The line was modified.

5. “Lines 291-295 paragraph belongs to Discussion.” Those lines were moved to Discussion.

6. “Lines 353-356 Paragraph needs to be rewritten.” The paragraph was rewritten.

Reviewer #2

1. “In the Results section, Transcriptomic analysis of the wild Chiltepin (CH; Capsicum annuum var. glabriusculum) and the cultivated Serrano Tampiqueño (ST; Capsicum annuum L. cv. ‘Tampiqueño 74’) was performed at two stages (Fig 1A), the immature stage at 20 days after anthesis (DAA) for both CH and ST, and at mature state at 60 DAA for ST and 68 DAA for CH [45]. I did not find that the fruit of ST

in Fig. 1A. Moreover, Please check if the reference [45] is appropriate.” Response: Fig. 1 was modified. Now panel A shows both accessions, and the DAA at which those were sampled are annotated. Reference [45] was removed.

Aditional changes:

1. Author’s name was changed from “A. Corina Hayano-Kanashiro” to “Corina Hayano-Kanashiro”.

2. Description of Figure 1 A) was changed to include both accessions over different development and

maturation stages.

3. Line 514: reference [107] was added to comply with KEGG copyright conditions.

4. Fig 6 was remade to improve resolution and readability.

Octavio Martínez Corresponding author 

Corina Hayano-Kanashiro Co-corresponding author

---

## [Decision Letter · Decision Letter 1]

4 Aug 2021

Transcriptomic analysis of a wild and a cultivated varieties of Capsicum annuum over fruit development and ripening

PONE-D-21-14082R1

Dear Dr. Martínez,

We’re pleased to inform you that your manuscript has been judged scientifically suitable for publication and will be formally accepted for publication once it meets all outstanding technical requirements.

Kind regards,

Xiang Jia Min, Ph. D.

Academic Editor

PLOS ONE

Additional Editor Comments (optional):

Reviewers' comments:

Reviewer's Responses to Questions

**Comments to the Author**

1. If the authors have adequately addressed your comments raised in a previous round of review and you feel that this manuscript is now acceptable for publication, you may indicate that here to bypass the “Comments to the Author” section, enter your conflict of interest statement in the “Confidential to Editor” section, and submit your "Accept" recommendation.

Reviewer #1: All comments have been addressed

Reviewer #2: All comments have been addressed

2. Is the manuscript technically sound, and do the data support the conclusions?

Reviewer #1: Yes

Reviewer #2: Yes

3. Has the statistical analysis been performed appropriately and rigorously? 

Reviewer #1: Yes

Reviewer #2: Yes

4. Have the authors made all data underlying the findings in their manuscript fully available?

Reviewer #1: Yes

Reviewer #2: Yes

5. Is the manuscript presented in an intelligible fashion and written in standard English?

Reviewer #1: Yes

Reviewer #2: Yes

6. Review Comments to the Author

Reviewer #1: (No Response)

Reviewer #2: The manuscript has been revised as requested. I think the manuscript is suitable for publication in PLOS ONE.

7. PLOS authors have the option to publish the peer review history of their article (what does this mean?). If published, this will include your full peer review and any attached files.

Reviewer #1: No

Reviewer #2: No

---

## [Editor Report · Acceptance letter]

6 Aug 2021

PONE-D-21-14082R1 

Transcriptomic analysis of a wild and a cultivated varieties of *Capsicum annuum* over fruit development and ripening 

Dear Dr. Martínez:

I'm pleased to inform you that your manuscript has been deemed suitable for publication in PLOS ONE. Congratulations! Your manuscript is now with our production department. 

Kind regards, 

on behalf of

Dr. Xiang Jia Min

Academic Editor

PLOS ONE